# Perceived stigma and depression among the HIV-positive adult people in Ethiopia: A systematic review and meta-analysis

Fikreab Desta[1]*, Demisu Zenbaba[1], Biniyam Sahiledengle[1], Yohannes Tekalegn[1], Demelash Woldeyohannes[2,3], Daniel Atlaw[4], Fikadu Nugusu[1], Lemlem Daniel Baffa[5], Degefa Gomora[6], Girma Beressa[1]

1 Department of Public Health, School of Health Sciences, Madda Walabu University, Robe, Oromia, Ethiopia, 2 School of Public Health, College Medicine and Health Sciences, Wachemo University, Hossana, Ethiopia, 3 Monash Centre for Health Research and Implementation, School of Public Health and Preventive Medicine, Monash University, Melbourne, Australia, 4 Biomedical Unit, School of Medicine, Madda Walabu University, Robe, Oromia, Ethiopia, 5 Department of Human Nutrition, University of Gondar, Gondar, Ethiopia, 6 Department of Midwifery, School of Health Sciences, Madda Walabu University, Robe, Oromia, Ethiopia

* fikerbuze@gmail.com

**Data Availability Statement:** All relevant data are within the paper and its Supporting Information files.

## Abstract

### Introduction

Depression is one of the most common psychiatric disorders, affecting approximately 60% of people infected with the human immunodeficiency virus (HIV). Low and middle-income countries (LMICs), including Ethiopia, bear a disproportionate burden of depression among HIV/AIDS patients. Several factors, including perceived stigma, have been linked to increased depression among HIV/AIDS patients. Therefore, we aimed to estimate the pooled effect of perceived stigma on depression among HIV/AIDS patients in Ethiopia.

### Methods

For this systematic review and meta-analysis, we systematically retrieved all relevant studies starting from January 1, 2000 to June 1, 2022 from PubMed, HINARI, and Google Scholar. This review included observational studies that reported the effect of perceived stigma on the prevalence of depression among HIV-positive adults in Ethiopia. The effect estimate of the pooled effect of perceived stigma on depression was conducted using DerSimonian-Laird's random effect model using STATA/MP version 16. An adjusted odds ratio (AOR), along with a 95% confidence interval (CI), was conducted to estimate the strength of the association between perceived stigma and depression.

### Results

Eleven studies with a total of 4,153 HIV-positive adults were included for meta-analysis. The results of the meta-analysis revealed that the odds of depression were higher among patients with perceived stigma (AOR: 3.78, 95% CI: 2.73, 5.24). The pooled prevalence of depression among HIV/AIDS patients in Ethiopia was 39% (95% CI: 32%, 46%) (I2 = 98%,

**Funding:** This research received no specific grant from any funding agency in the public, commercial or not-for-profit sectors.

**Competing interests:** The authors have declared that no competing interests exist.

$p \leq 0.0001$). The subgroup analysis revealed that the primary studies conducted in the Oromia region had the highest pooled prevalence of depression at 48% (95% CI: 32%, 63%).

## Conclusion

The pooled estimates of the meta-analysis revealed that perceived stigma and depression were strongly associated. Stigma and depression screenings should be carried out for additional treatments and prevention, and programs supporting Ethiopia's PLWHA population should be strengthened.

## Introduction

Globally, 37.7 million people are living with human immune deficiency virus and acquired immune deficiency syndrome (HIV/AIDS) and more than half of them are in Africa [1]. Mental health is inextricably linked to communicable diseases like HIV/AIDS [2]. Neuropsychiatric disorders account for approximately 14% of the global disease burden, owing primarily to the chronically disabling nature of depression [3]. The majority of the disease's burden is concentrated in LMICs. Depression affects approximately 60% of HIV-positive people; However, half of all people living with HIV with depression go undiagnosed and untreated [4]. By 2030, depression and HIV/AIDS are expected to be the world's two leading causes of disability [5].

The World Health Organization estimated that depression will have the dubious distinction of having the second-highest global disease burden after 20 years [6]. Most settings, including sub-Saharan Africa (SSA), now view HIV infection as a chronic illness [7]. Insufficient focus has been placed on mental health conditions among the non-communicable disorders seen in PLWHIV on ART, especially in SSA, where the majority of people live with HIV [8].

Stigma is a negative attitude and abuse directed at people living with HIV/AIDS [9]. HIV/AIDS stigma is prejudice, discounting, ridiculing, and discrimination aimed against those who are suspected of having AIDS or HIV [10, 11]. Perceived stigma describes how people living with HIV feel when they are treated unfavorably because of their HIV status by partners, family, friends, health care providers, and members of their community [11, 12]. Stigma has also been linked to how much people living with HIV believe their community stigmatizes them [9, 13].

Evidence on the association of perceived stigma with depression is critical in overcoming the negative impact of depression among HIV/AIDS patients. Despite this, the mental health of the HIV-positive population in general, and depression in particular, has received inadequate attention. Even though, those studies are single studies they may not be robust enough to be generalizable. Hence, the aim of this systematic review and meta-analysis was to estimate the pooled effects of perceived stigma on depression among HIV/AIDS patients in Ethiopia. This meta-analysis finding would assist decision-makers and other mental health stakeholders in reducing the magnitude of depression and the associated consequences among HIV/AIDS patients by implementing effective interventions.

## Methods

### The protocol and registration

We conducted this systematic review and meta-analysis based on the Preferred Reporting Items for Systematic Review and Meta-Analysis (PRISMA) statement guideline [14]. The

protocol for this review in the process for registration on International Prospective Register of Systematic Reviews (PROSPERO). The systematic review was carried out in accordance to the Joanna Briggs Institute methodology for systematic reviews of association evidence [15].

## Search strategies

A comprehensive search strategy was conducted on depressive symptoms and associated factors among HIV/AIDS patients to identify all relevant studies. A systematic literature search for the relevant papers was carried out in PubMed, Hinari, and Google Scholar. The search was restricted to papers published starting from January 1, 2000 to June 1, 2022 in Ethiopia and published in English. The population, etiology/risk, and outcome (PEO) format was used to search the relevant studies using the Depression [Mesh] OR perceived stigma OR "depressive symptoms [Mesh]" AND Ethiopia. "Perceived stigma", "Depression", Depressive symptoms, "Effect", "HIV", "AIDS", and "Ethiopia" were combinations of relevant keywords used.

## Eligibility criteria

**Inclusion criteria.** All observational studies conducted on the prevalence of depression; studies that assessed the association of perceived stigma with depression among HIV positive patients in Ethiopia; studies published and accessible before June 1, 2022; articles written in English; and citations with abstract and/or full text were eligible for the current systematic review and meta-analysis.

**Exclusion criteria.** Articles which were not fully accessed because of the inability to assess the quality of articles in the absence of full text, duplicate reports, systematic reviews and meta-analyses, qualitative studies, and inconsistent outcome measures were excluded from the review.

## Types of studies

We included studies reported on the institution-based prevalence of depression and the association between perceived stigma and depression in Ethiopia. Moreover, this study included all full-text papers written in English, institution-based design, and published prior to June 1, 2022.

## Exposure of interests

Perceived stigma was assessed using the 11-item or 12-item HIV stigma scale, with individuals who scored greater than or equal to the mean regarded as having stigma throughout the data extraction process.

## Study selection and data collection

All the studies reviewed through different electronic data bases were collected and uploaded into Endnote Version X.8 (Taylor and Francis) software and duplicates were removed. Full-text papers were manually downloaded using Endnote software. The eligibility of each study was assessed independently by two reviewers (FD. & BS.). Any disagreement that arose between the reviewers was resolved through discussion mediated with the additional reviewers (YT and LD.).

## Assessment of the quality of the individual studies

We used Joanna Briggs Institutes' (JBI) critical appraisal checklist for cross-sectional studies. The total is composed of eight parameters. 1. Were the criteria for inclusion in the sample

clearly defined? 2. Were the study subjects and the setting described in detail? 3. Was the exposure measured in a valid and reliable way? 4. Were objective, standard criteria used for the measurement of the condition? 5. Were confounding factors identified? 6. Were strategies to deal with confounding factors stated? 7. Were the outcomes measured in a valid and reliable way? 8. Was appropriate statistical analysis used? three authors (DG, LD and DW) evaluated the risk of bias of the full text considered to be included in the meta-analysis. The overall risk of bias was then scored according to the number of high risks of bias per study: low ($\leq$2), moderate (3–4), and high ($\geq$ 5) (**S1 File**).

## Outcome assessments

We determined the association between perceived stigma and depression in the form of the adjusted odds ratio (AOR). The outcome variable of interest was depression and was defined as the presence of depressed mood, loss of interest or pleasure, decreased energy, feelings of guilt or low self-worth, disturbed sleep or appetite, and poor concentration.

## Data extraction and management

Two authors (FN. & GB.) independently reviewed all essential parameters extracted from each study using Microsoft Excel. Disagreements between the two authors were resolved through consensus with additional authors (DZ. & DA.). The data extraction form was prepared with the help of the data extraction tool for prevalence studies developed by the Joanna Briggs Institute (JBI). Authors, years of publication, study area or region, study design, response rate, sample size, the prevalence of depression, mean age, and adjusted odds ratio, with their confidence interval (CI) were extracted for each study.

## Data synthesis

The extracted data were exported to Stata/MP 16 software. The random effect model using the DerSimonian-Liard model was used to estimate the pooled effect of perceived stigma and the prevalence of depression. Effect sizes were expressed as odds ratio (OR) and percentages along with a 95% confidence interval (CI). Heterogeneity in meta-analysis is mostly inevitable due to differences in study quality, sample size, method, and different outcome measurements across studies.

The presence and degree of heterogeneity were assessed statistically using the Cochrane Q-test (p-value <0.1) and $I^2$ statistics, respectively. A subgroup analysis was carried out based on the study area and the tool used to indicate the variation of estimated points between primary studies. In addition, a sensitivity analysis was performed to examine the impact of individual studies on the pooled values. A random-effects model was used to conduct a univariate meta-regression using the study area. A funnel plot and Egger's statistical test were used to assess publication bias (small study effect). The Egger's regression test at a (P-value of <0.05) was deemed evidence of publication bias. The statistical significance of the association was declared at a p-value of less than 0.05.

**Ethics approval.** This study does not involve human participants. The systematic review and meta-analysis use primary data synthesis; hence, ethics approval is not required.

## Results

A total of 1383 studies were identified through the search of electronic databases (PubMed/Medline, Hinari, and Google Scholar). Three other articles were identified through reference tracing and from registries. After omitting 69 duplicates through Endnote reference manager

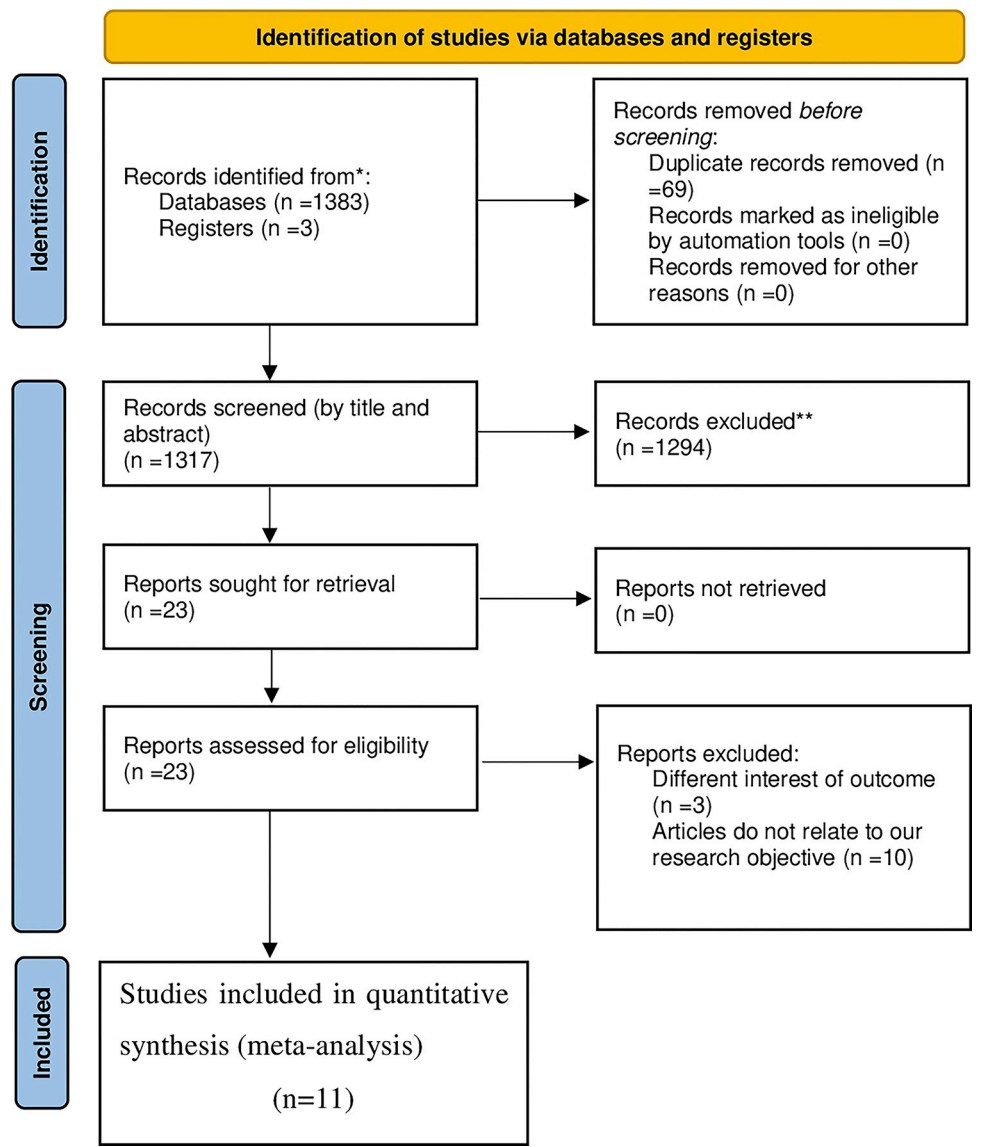

**Fig 1. PRISMA flow diagram of article selection for systematic review and meta-analysis of the pooled adjusted odds ratio of perceived stigma and depression in Ethiopia [27].**

and manual tracing, a total of 1317 records were screened using their titles and abstracts. Then, a full-text assessment of 23 potentially relevant articles resulted in 11 studies [16–26] that met the eligibility criteria and quality assessment and were hence included in the systematic review and meta-analysis (**Fig 1**).

## Study characteristics

A total of 11 studies with a total of 4153 study participants were included in this systematic review and meta-analysis. Eleven of the included studies [16–26] were used cross-sectional study design and were conducted in institution. The effect of the perceived stigma on depression among HIV patients varied as (AOR: 1.6, 95% CI: 1.04, 2.67) in study conducted in southern nation nationalities and people's region (SNNPR) [19] and (AOR: 10.2, 95% CI: 4.26, 24.4) in a study conducted Amhara region [21] (**Table 1**).

**Table 1. Characteristics of included studies in systematic review and meta-analysis of the effect of perceived stigma on depression in Ethiopia (n = 11).**

| Primary author | year of publication | study area | study design | sample size | response rate (%) | Tools | Prevalence of depression (%) | Mean age ±SD |
|---|---|---|---|---|---|---|---|---|
| Abadiga et al. [16]. | 2019 | Oromia | Cross-sectional | 393 | 97.30 | PHQ-9 | 41.70 | 25.6±9.45 |
| Abdisa et al. [17]. | 2021 | Oromia | Cross-sectional | 384 | 90.14 | PHQ-9 | 42.96 | 36.83±10.68 |
| Amha et al. [18]. | 2022 | Amhara | Cross-sectional | 266 | 97.40 | PHQ-9 | 39.10 | not reported |
| Beyamo et al. [19]. | 2020 | SNNPR | Cross-sectional | 410 | 98.30 | PHQ-9 | 50.50 | 33.05±9.34 |
| Dagne et al. [20]. | 2018 | Amhara | Cross-sectional | 416 | not reported | PHQ-9 | 38.94 | 38 |
| Damtie et al. [21]. | 2021 | Amhara | Cross-sectional | 380 | 97.90 | PHQ-9 | 15.50 | not reported |
| Duko et al. [22]. | 2018 | SNNPR | Cross-sectional | 401 | 96.20% | PHQ-9 | 48.60% | 38± 10.228 |
| Duko et al. [23]. | 2019 | SNNPR | Cross-sectional | 363 | 100.00% | HADS | 32.00% | 37.66±10.03 |
| Seid et al. [24]. | 2020 | Amhara | Cross-sectional | 395 | 93.50% | PHQ-9 | 20.00% | 38 median |
| Tesfaw et al. [25]. | 2015 | Addis Ababa | Cross-sectional | 417 | 100.00% | HADS | 41.20% | 37.44±10.07 |
| Workye et al. [26]. | 2018 | SNNPR | Cross-sectional | 328 | 96.50% | PHQ-9 | 37.50% | 37.6 |

SNNPR: southern nation nationalities and people's region, PHQ-9: patient health questionnaire-9; HADS: Hospital Anxiety and Depression Scale; SD: standard deviation.

## Meta-analysis

### The perceived stigma and depression

Eleven cross-sectional studies were included in this meta-analysis to determine the effect of perceived stigma on depression. The studies exhibited significant heterogeneity ($I^2$ = 65.4%, P < 0.001). Hence, a random-effect meta-analysis model was used to estimate the pooled adjusted OR. This meta-analysis revealed that perceived stigma is significantly associated with depression. The likelihood of developing depression was higher among people living with HIV (AOR = 3.78; 95% CI: 2.73, 5.24) (**Fig 2**).

### Subgroup analysis and publication bias

A subgroup analysis was conducted based on the study area or region. The result of the sub-group analysis revealed that the primary studies that were conducted in the Oromia region had the highest pooled estimate of the prevalence of depression at 48% (95% CI: 32%, 63%) (**Fig 3**). Funnel plots of standard error with effect size were used to assess publication bias (**Fig 4**). The Egger test showed that there was evidence of a small study effect (P-value = 0.014).

### Sensitivity analysis for the perceived stigma and depression

To estimate the effect of each individual study on the pooled estimate of the association between perceived stigma and depression, we performed a sensitivity analysis using the rando-meffect model. Based on the results, a single study has no significant effect on the pooled esti-mate. The pooled estimated for adjusted AOR ranged between 3.5 (2.53–4.71) and 4 (3.03–5.58) after omitting each study (**Fig 5**).

## Discussion

This systematic review and meta-analysis aimed to estimate the pooled effect of perceived stigma on depression among HIV/AIDS patients in Ethiopia. According to the eleven studies analyzed, perceived stigma had a statistically significant effect on depression among people liv-ing with HIV/AIDS.

The odd of having depression among people living with HIV/AIDS who had perceived stigma were more than three times higher than their counterparts. The positive association

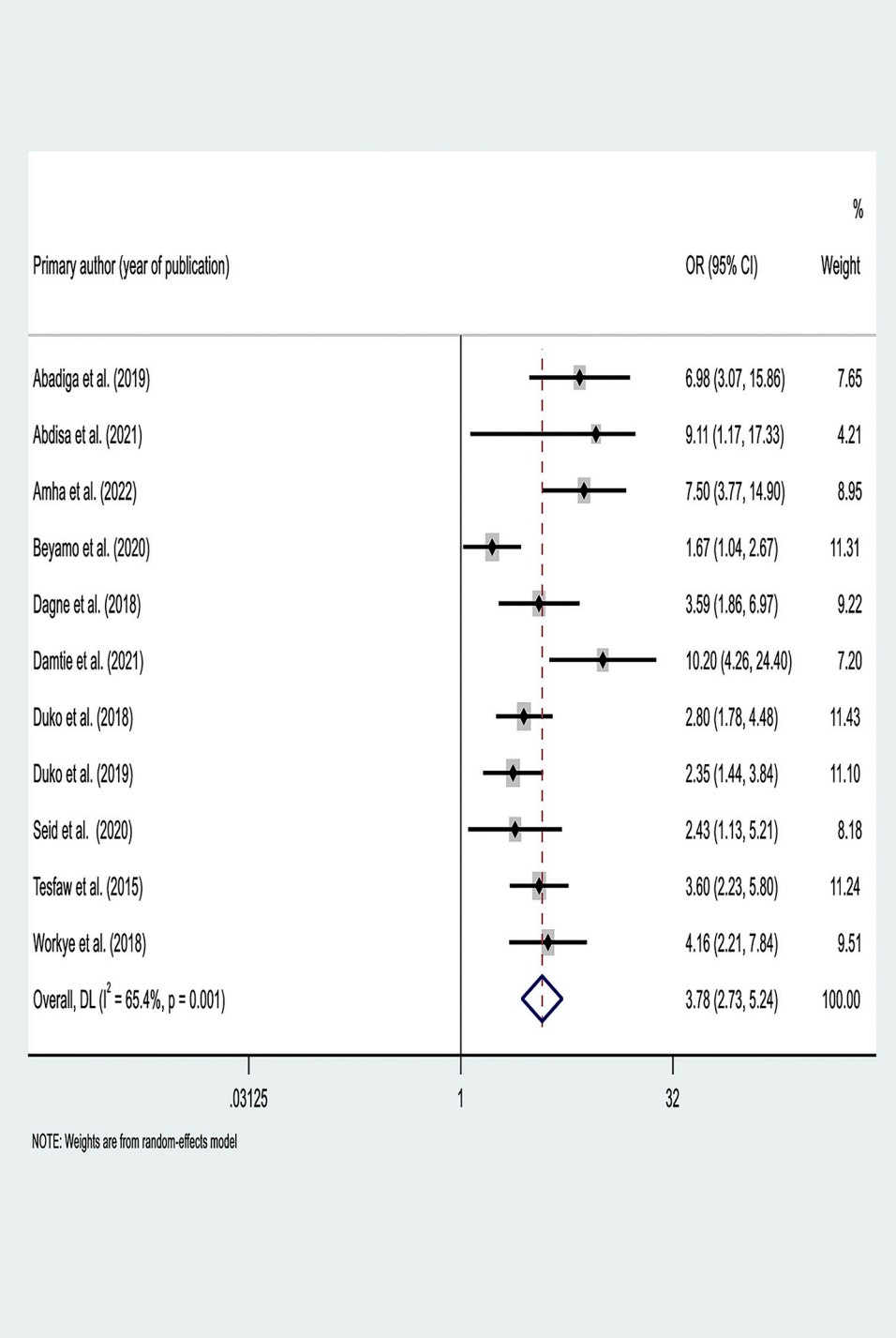

**Fig 2. Forest plot of pooled adjusted OR between perceived stigma and depression among HIV/AIDS patients in Ethiopia.**

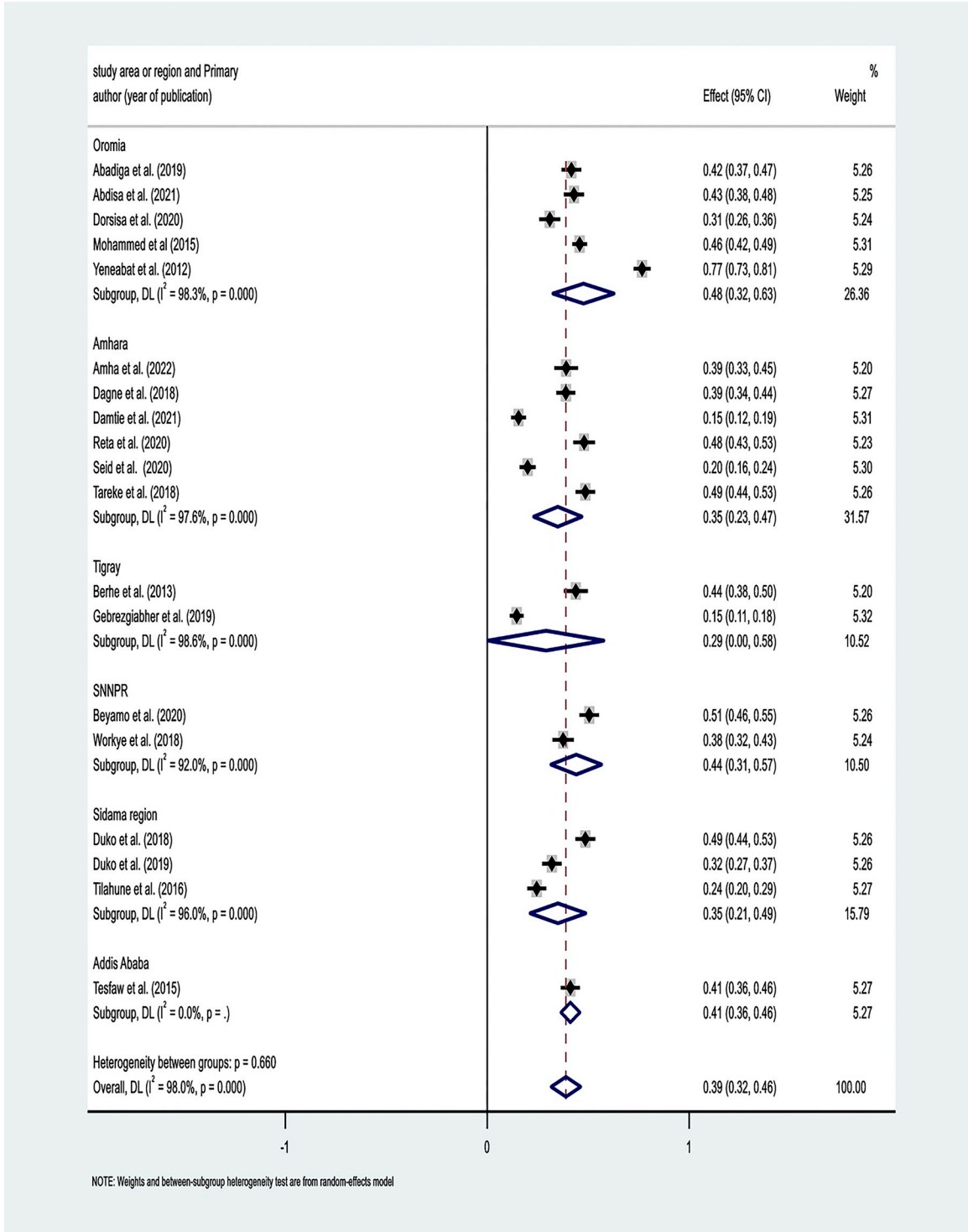

**Fig 3. The pooled estimate of prevalence of depression among HIV/AIDS patients in Ethiopia based on study area/region, 2022.**

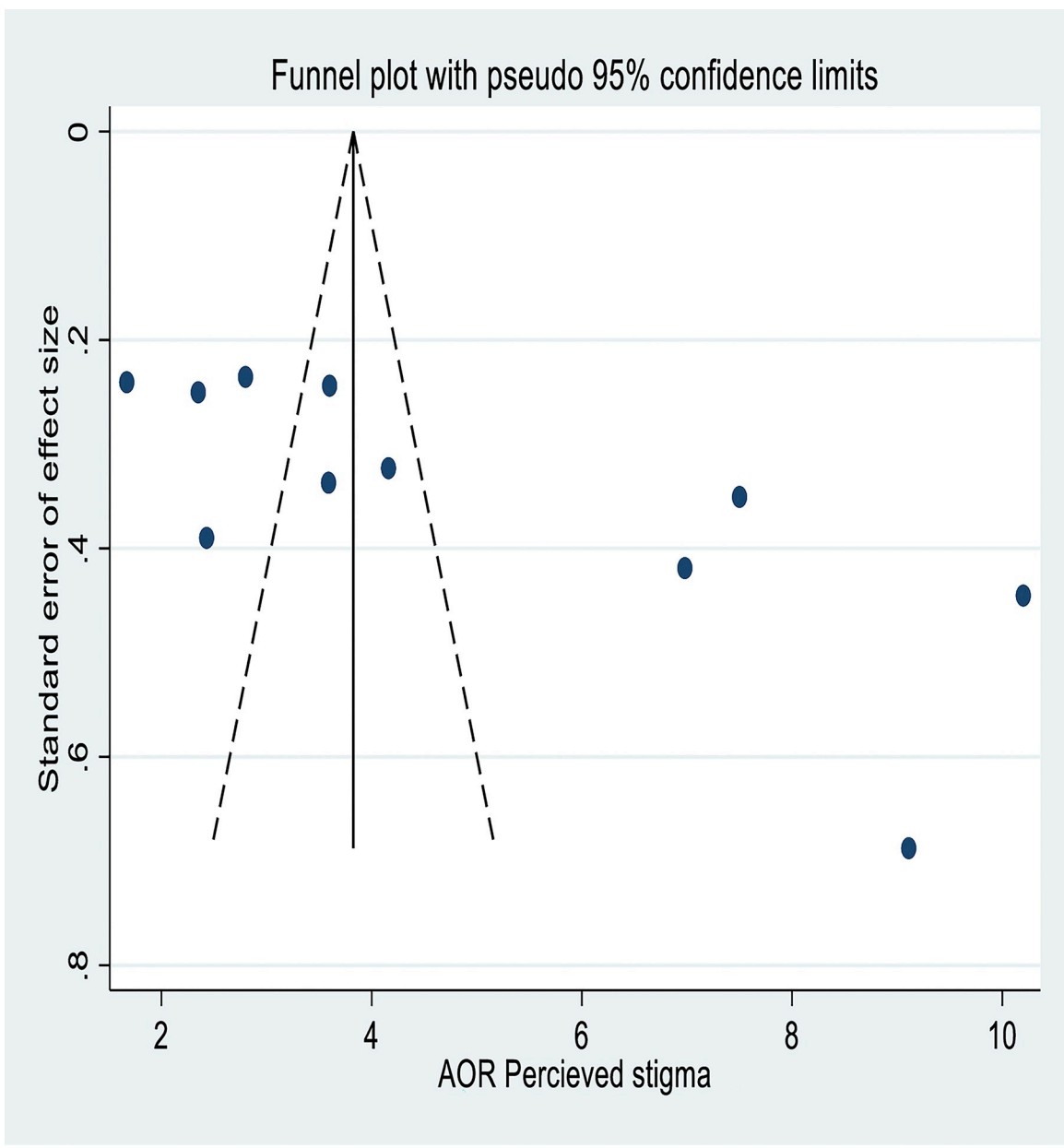

**Fig 4. A funnel plot for assessing publication bias, 2022.**

between perceived stigma and depression in the current study was consistent with a finding from a low-income country [28]. The possible reason could be that patients feel treated as unfavorably because of their HIV status by others, and those who did not share their problems with others experienced stress and social isolation, which can have a negative impact on their mental well-being [2]. Stigma may increase their isolation and loneliness, which might also contribute to the positive relationship between perceived stigma and depression.

A total of nineteen studies that assessed the prevalence of depression in Ethiopian HIV/AIDS patients were included in the meta-analysis. The pooled estimate of the prevalence of depression in this study was 39%. This was significantly higher than Ethiopia's general population's estimated pooled prevalence of depression, which ranged from 9.1% [29] to 11% [30].

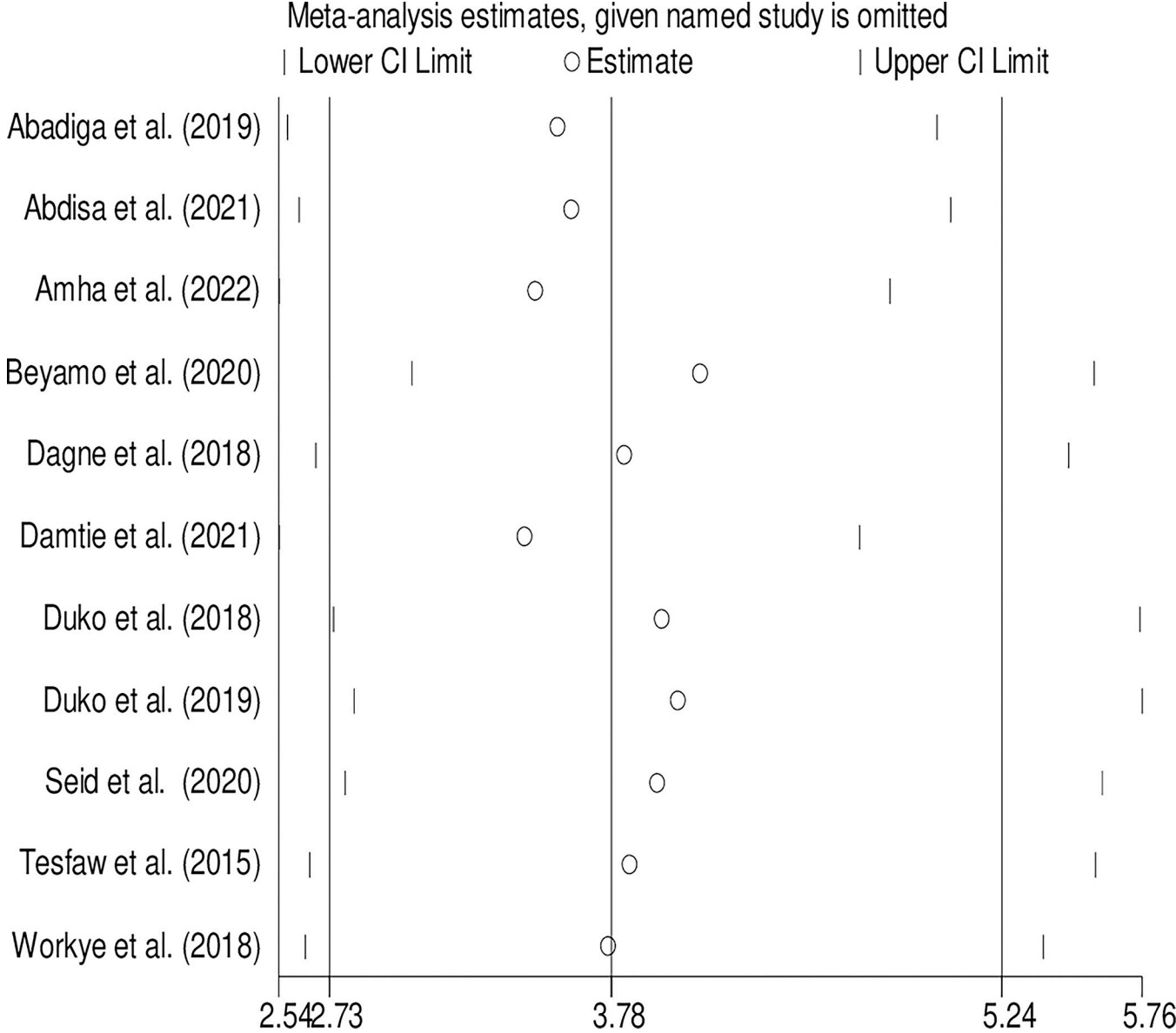

**Fig 5. Sensitivity analysis for the pooled adjusted odds ratio of the perceived stigma and depression.**

This indicates that the health of those people living with HIV/AIDS was significantly impacted by depression. On the other hand, this finding of meta-analysis is consistent with the findings of meta-analysis [2, 31] in which the average prevalence of depression among HIV/AIDS patients were 38% and 38.93%, respectively.

We conducted the subgroup analysis by the study area or region, and the highest prevalence of depression observed was 48% in primary studies that were conducted in the Oromia region and the lowest was observed at 29% in the Tigray region. The discrepancy might be due to the difference in number of the original studies included in the subgroup analysis. Only two primary studies were conducted in the Tigray region to assess depression, whereas five studies were conducted in the Oromia region, potentially increasing the precision of the pooled estimate for depression.

## Limitation of the meta analysis

This study aimed to examine pooled effect of perceived stigma on depression with no prior evidence on the condition. Although we strictly followed the PRISMA checklist during the conduct of this review, which enhances its quality for the readers; the finding must be interpreted in light of the following limitation; it might be lacking national representativeness since primary studies were found only from six administrative regions, namely, Addis Ababa, Amhara region, Oromia region, Sidama region, Tigray and Southern Nations nationalities people's (SNNP) region; this could bias the estimated prevalence of depression for the entire Ethiopian context.

## Conclusion

The meta-analysis found a strong pooled effect of depression on perceived stigma, as well as a high pooled prevalence of depression among HIV patients. HIV/AIDS patients with perceived stigma were more likely to suffer from depression. Hence, we recommend the inclusion and strengthening of mental health and psychosocial services in routine HIV/AIDS care. Stigma and depression screenings should be carried out for additional treatments and prevention, and programs supporting Ethiopia's PLWHA population should be strengthened.

## Supporting information

**S1 Checklist.**
(DOCX)

**S1 File.**
(XLSX)

## Author Contributions

**Conceptualization:** Fikreab Desta, Demisu Zenbaba, Biniyam Sahiledengle, Yohannes Tekalegn, Demelash Woldeyohannes, Daniel Atlaw, Fikadu Nugusu, Lemlem Daniel Baffa, Degefa Gomora, Girma Beressa.

**Data curation:** Fikreab Desta, Biniyam Sahiledengle, Yohannes Tekalegn, Daniel Atlaw, Fikadu Nugusu.

**Formal analysis:** Fikreab Desta, Demisu Zenbaba, Biniyam Sahiledengle, Demelash Woldeyohannes, Daniel Atlaw, Lemlem Daniel Baffa, Degefa Gomora, Girma Beressa.

**Investigation:** Fikreab Desta, Demisu Zenbaba, Biniyam Sahiledengle, Yohannes Tekalegn, Daniel Atlaw, Fikadu Nugusu, Lemlem Daniel Baffa.

**Methodology:** Fikreab Desta, Demisu Zenbaba, Biniyam Sahiledengle, Demelash Woldeyohannes, Daniel Atlaw, Fikadu Nugusu, Lemlem Daniel Baffa, Degefa Gomora, Girma Beressa.

**Software:** Fikreab Desta, Demisu Zenbaba, Biniyam Sahiledengle, Yohannes Tekalegn, Demelash Woldeyohannes, Daniel Atlaw, Fikadu Nugusu, Lemlem Daniel Baffa, Degefa Gomora.

**Supervision:** Fikreab Desta, Demisu Zenbaba, Biniyam Sahiledengle, Fikadu Nugusu, Girma Beressa.

**Validation:** Fikreab Desta, Demisu Zenbaba, Biniyam Sahiledengle, Fikadu Nugusu, Degefa Gomora.

**Visualization:** Fikreab Desta, Yohannes Tekalegn, Demelash Woldeyohannes, Lemlem Daniel Baffa, Girma Beressa.

**Writing – review & editing:** Fikreab Desta, Yohannes Tekalegn, Degefa Gomora, Girma Beressa.

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
