## [Decision Letter · Decision Letter 0]

20 Dec 2023

PONE-D-22-30518Perceived stigma and depression among the HIV-positive adult people in Ethiopia: a systematic review and meta-analysisPLOS ONE

Dear Dr. Desta,

Thank you for submitting your manuscript to PLOS ONE. After careful consideration, we feel that it has merit but does not fully meet PLOS ONE’s publication criteria as it currently stands. Therefore, we invite you to submit a revised version of the manuscript that addresses the points raised during the review process.

We look forward to receiving your revised manuscript.

Kind regards,

Humayun Kabir

Academic Editor

PLOS ONE

Journal Requirements:

2. Thank you for submitting the above manuscript to PLOS ONE. During our internal evaluation of the manuscript, we found significant text overlap between your submission and previous work in the Methods and Conclusions.

Please revise the manuscript to rephrase the duplicated text, cite your sources, and provide details as to how the current manuscript advances on previous work. Please note that further consideration is dependent on the submission of a manuscript that addresses these concerns about the overlap in text with published work.

We will carefully review your manuscript upon resubmission and further consideration of the manuscript is dependent on the text overlap being addressed in full. Please ensure that your revision is thorough as failure to address the concerns to our satisfaction may result in your submission not being considered further.

“None declared”

Reviewers' comments:

Reviewer's Responses to Questions

**Comments to the Author**

1. Is the manuscript technically sound, and do the data support the conclusions?

Reviewer #1: Yes

Reviewer #2: Yes

2. Has the statistical analysis been performed appropriately and rigorously? 

Reviewer #1: Yes

Reviewer #2: Yes

3. Have the authors made all data underlying the findings in their manuscript fully available?

Reviewer #1: Yes

Reviewer #2: Yes

4. Is the manuscript presented in an intelligible fashion and written in standard English?

Reviewer #1: No

Reviewer #2: Yes

5. Review Comments to the Author

Reviewer #1: Review for Plose one

Title: Perceived stigma and depression among the HIV-positive adult people in Ethiopia: a Systematic review and meta-analysis

Manuscript number: PONE-D-22-30518

Thanks, dear editor for your invitation to review this work.

1. The abstract section needs intensive revision and the recommendations should be based on the findings. The abstract seems grammatically problem and I think the software STATA 16 version is not done currently how to use this software and Please rewrite the abstract section based on Plose journal format

2. Introduction:

3. Result

Just focus on Pooled prevalence not sample size only

Neither the figure nor the table were watched.

4. Conclusion

The conclusion should clearly show the major findings and the authors are expected to recommend based on modifieable variables.

Reviewer #2: The manuscript is well-written and follows the systematic review and meta-analysis guidelines.

I do have a few comments:

1. I would advise the authors to specify a specific start year for their search instead of saying "from inception."

2. Which institutions are the authors referring to?

3. In Table 1, try to be consistent by either reporting the numbers to the nearest tens or hundreds.

6. PLOS authors have the option to publish the peer review history of their article (what does this mean?). If published, this will include your full peer review and any attached files.

Reviewer #1: No

Reviewer #2: **Yes: **Mona Abdelrehim

---

## [Author Response · Author response to Decision Letter 0]

1 Jan 2024

thank you both reviewers for your time and valuable constructing comments and I have noted every reviewers concerns on the response documents for now I have no any comments regarding reviewers. with kind regards!!

---

## [Decision Letter · Decision Letter 1]

16 Apr 2024

Perceived stigma and depression among the HIV-positive adult people in Ethiopia: a systematic review and meta-analysis

PONE-D-22-30518R1

Dear Dr. Desta,

We’re pleased to inform you that your manuscript has been judged scientifically suitable for publication and will be formally accepted for publication once it meets all outstanding technical requirements.

Kind regards,

Matthew J. Mimiaga, ScD, MPH

Academic Editor

PLOS ONE

Additional Editor Comments (optional):

The author's nicely addressed all concerns raised by the reviewers.

Reviewers' comments:

Reviewer's Responses to Questions

**Comments to the Author**

1. If the authors have adequately addressed your comments raised in a previous round of review and you feel that this manuscript is now acceptable for publication, you may indicate that here to bypass the “Comments to the Author” section, enter your conflict of interest statement in the “Confidential to Editor” section, and submit your "Accept" recommendation.

Reviewer #2: All comments have been addressed

2. Is the manuscript technically sound, and do the data support the conclusions?

Reviewer #2: Yes

3. Has the statistical analysis been performed appropriately and rigorously? 

Reviewer #2: Yes

4. Have the authors made all data underlying the findings in their manuscript fully available?

Reviewer #2: (No Response)

5. Is the manuscript presented in an intelligible fashion and written in standard English?

Reviewer #2: Yes

6. Review Comments to the Author

Reviewer #2: (No Response)

7. PLOS authors have the option to publish the peer review history of their article (what does this mean?). If published, this will include your full peer review and any attached files.

Reviewer #2: **Yes: **Mona Abdelrehim

---

## [Editor Report · Acceptance letter]

23 May 2024

PONE-D-22-30518R1 

PLOS ONE

Dear Dr. Desta, 

I'm pleased to inform you that your manuscript has been deemed suitable for publication in PLOS ONE. Congratulations! Your manuscript is now being handed over to our production team.

Kind regards, 

on behalf of

Dr. Matthew J. Mimiaga 

Academic Editor

PLOS ONE